# Vitamin K2 (MK-7) attenuates LPS-induced acute lung injury via inhibiting inflammation, apoptosis, and ferroptosis

Yulian Wang[1⊛], Weidong Yang[1⊛], Lulu Liu[1], Lihong Liu[1], Jiepeng Chen[2], Lili Duan[2], Yuyuan Li[ID][3]*, Shuzhuang Li[1]*

**1** College of Basic Medical Science, Dalian Medical University, Dalian, China, **2** Sungen Bioscience Co., Ltd., Guangdong, China, **3** Advanced Institute for Medical Sciences, Dalian Medical University, Dalian, China

⊛ These authors contributed equally to this work.
* shuzhuangli@126.com (SL); liyuyuan831221@163.com (YL)

**Data Availability Statement:** All relevant data are within the paper.

**Funding:** This work was supported by the China Health Promotion Association (Z093001to

## Abstract

Acute lung injury (ALI) is a life-threatening disease that has received considerable critical attention in the field of intensive care. This study aimed to explore the role and mechanism of vitamin K2 (VK2) in ALI. Intraperitoneal injection of 7 mg/kg LPS was used to induce ALI in mice, and VK2 injection was intragastrically administered with the dose of 0.2 and 15 mg/kg. We found that VK2 improved the pulmonary pathology, reduced myeloperoxidase (MPO) activity and levels of TNF-α and IL-6, and boosted the level of IL-10 of mice with ALI. Moreover, VK2 played a significant part in apoptosis by downregulating and upregulating Caspase-3 and Bcl-2 expressions, respectively. As for further mechanism exploration, we found that VK2 inhibited P38 MAPK signaling. Our results also showed that VK2 inhibited ferroptosis, which manifested by reducing malondialdehyde (MDA) and iron levels, increasing glutathione (GSH) level, and upregulated and downregulated glutathione peroxidase 4 (GPX4) and heme oxygenase-1 (HO-1) expressions, respectively. In addition, VK2 also inhibited elastin degradation by reducing levels of uncarboxylated matrix Gla protein (uc-MGP) and desmosine (DES). Overall, VK2 robustly alleviated ALI by inhibiting LPS-induced inflammation, apoptosis, ferroptosis, and elastin degradation, making it a potential novel therapeutic candidate for ALI.

## Introduction

Acute lung injury (ALI) and its severe form, acute respiratory distress syndrome (ARDS), are severe respiratory system disease [1]. ALI is characterized by disruption of the lung capillary endothelium and alveolar-capillary membrane barrier, overactivated inflammatory response, and pulmonary edema [2]. Lipopolysaccharide (LPS) is the principal component of the outer membranes of gram-negative bacteria, which can trigger strong inflammatory responses and cause severe lung injury. The molecular process of ALI is believed to involve complex interactions between inflammation, oxidative stress, apoptosis, and ferroptosis [3, 4]. Cytokine storms are a result of the markedly elevated proinflammatory cytokines such as tumor necrosis factor-

Shuzhuang Li) and the Nature Science Foundation of Liaoning Province, China (2019-ZD-0648 to Yuyuan Li).

**Competing interests:** The authors have declared that no competing interests exist.

**Abbreviations:** ALI, acute lung injury; ARDS, acute respiratory distress syndrome; COVID-19, Coronavirus Disease 2019; DES, desmosine; DEX, dexamethasone; GPX4, glutathione peroxidase 4; GSH, glutathione; HE, hematoxylin / eosin; HO-1, heme oxygenase-1; IL-1β, interleukin-1β; IL-6, interleukin-6; IL-10, interleukin-10; LPS, lipopolysaccharide; MAPKs, mitogen-activated protein kinases; MDA, malondialdehyde; MPO, myeloperoxidase; PVDF, polyvinulidene difluoride; ROS, reactive oxygen species; SDS, sodium dodecyl sulfate-polyacrylamide gelelectrophoresis; SPF, specific-pathogen-free; TNF-α, tumor necrosis factor-α; uc-MGP, uncarboxylated matrix Gla protein; VK2, vitamin K2.

α (TNF-α), interleukin-1β (IL-1β), and interleukin-6 (IL-6), which have been proved to be major factors promoting the development of ALI [5]. These cytokines act on leukocytes to activate positive feedback of proinflammatory signals [6, 7]. At the same time, a large number of leukocytes' influx into the lungs also induces the production of reactive oxygen species (ROS) in hepatocytes and endoplasmic reticulum stress, leading to hepatocyte apoptosis and ferroptosis [8, 9]. Therefore, inhibiting inflammation, oxidative stress, apoptosis, and ferroptosis is crucial for alleviating LPS-induced ALI.

However, currently, there are no efficient drugs or therapies for LPS-induced ALI in clinical practice. A promising strategy to prevent and treat ALI is to suppress the excessive production of inflammatory cytokines through anti-inflammatory drugs, including corticosteroids, such as methylprednisolone [10], dexamethasone (DEX) [11], prednisolone [12]. However, the clinical application of corticosteroids is usually accompanied by various severe side effects, such as hyperglycemia, hypertension, hypokalemia, dyslipidemia, osteoporosis, myopathy and immunosuppression [13–15]. Therefore, it is necessary to find new drugs with better efficacy and safety for the treatment of ALI patients.

Vitamin K (VK) is a family of lipid-soluble molecules that exists mainly two forms: vitamin K1 (phylloquinone) and vitamin K2 (menaquinone). VK2, essential for human metabolism and health is predominantly present in cheese and fermented soybeans (natto) [16]. In recent years, VK2 has become a research hotspot for scientists due to its multiple pharmacological activities, including anti-inflammatory, anti-oxidant, anti-apoptosis, and anti-ferroptosis [17–19]. However, whether VK2 has a beneficial effect on LPS-induced ALI has not been investigated. Thereby, this study was conducted to evaluate if VK2 administration could offer protection against ALI/ARDS.

## Materials and methods

### Reagents

VK2 was provided by Sungen Bioscience Co., Ltd. (Guangdong, China). LPS (from Escherichia coli (055:B5)) and DEX were purchased from Sigma-Aldrich (St. Louis, MO, United States). Mouse myeloperoxidase (MPO) determination kit was purchased from the Jiancheng Bioengineering Institute of Nanjing (Nanjing, Jiangsu Province, China). ELISA kits for mouse TNF-α, IL-6 and IL-10 were purchased from Jiangsu Meimian Industrial Co., Ltd. (Jiangsu, China) and for uncarboxylated MGP (uc-MGP) and desmosine (DES) were from Shanghai Lengton Bioscience Co., Ltd. (shanghai, china). The malondialdehyde (MDA), glutathione (GSH), and Iron determination kits were purchased from Solarbio (Beijing, China). The antibodies against GAPDH, iNOS, and Caspase-3 were obtained from Abcame (Cambridge, MA). The antibodies against TLR4 and IL-6 were purchased from Wanleibio (Liaoning, China). The antibodies against P38, p-P38, and Bcl-2 were from Proteintech (Wuhan, China). The antibodies against glutathione peroxidase 4 (GPX4) and heme oxygenase-1 (HO-1) were obtained from Bioworld (Minnesota, USA). Other chemicals were conformed from reagent grade.

### Animal experiments

Specific-pathogen-free (SPF) male C57BL/6 mice (8–10 weeks old, 20–24 g body weight) were purchased from Liaoning Changsheng Biotechnology Co., Ltd. (Liaoning, China). The mice were kept in cages with food and water ad libitum during the whole period. The experiment was conducted with the approval of the Dalian Medical University Animal Care and Use Committee (AEE20051).

LPS (7 mg/kg) was dissolved in saline and administrated by intraperitoneal injection to mice and the acute lung injury model was established by LPS administration for 3 days. The

mice were randomly divided into six groups (7 mice per group): control group, LPS group, negative control group (LPS + Oil [vehicle]), positive control group (LPS + DEX), and the VK2 pretreatment groups at two different concentrations. VK2 was dissolved in soybean oil at different concentrations (0.2 and 15 mg/kg). Before LPS treatment, the VK2 pretreatment groups were pre-administered intragastrically for 6 days, while the control and model groups were given the same volume of solvent solution. In the positive control group, mice were Caspase-3. The treatment of mice was shown in **Fig 1A**. After deeply anesthetized with tribromoethanol, the lung and serum were collected from each mouse for further experiments.

## Histological evaluation of lung

Lung tissue samples were fixed in 4% (w/v) paraformaldehyde and then dehydrated, embedded in paraffin and stained with hematoxylin / eosin (HE). The severity of lung damage was semi-quantitatively scored as previously described [20].

## Determination of myeloperoxidase (MPO) activity and the content of ferroptosis biomarkers in lung tissue

Lung homogenate (10%, w/v), obtained by homogenizing the lung tissue and normal saline, was used for the measurements of MPO activities, and MDA, GSH and Iron content using test kits in accordance with the manufacturer's instructions.

## Measurement of inflammatory cytokines in serum

The serum was collected and the levels of pro-inflammatory cytokines TNF-α and IL-6, and anti-inflammatory cytokines IL-10 in the serum were measured by ELISA kits according to the manufacturer's recommendations.

## Assessment of the degradation level of elastic fibers

To evaluate the level of elastic fiber degradation, the concentrations of uc-MGP in serum and DES in lung tissue were quantified using ELISA. All measurements were carried out following manufacturer's instructions.

## Western blotting analysis

The lung tissues were lysed with RIPA Lysis Buffer to extract the proteins that were quantified by the BCA assay. Denatured proteins were separated using sodium dodecyl sulfate-polyacrylamide gelelectrophoresis (SDS-PAGE) and then transferred onto polyvinulidene difluoride (PVDF) membranes. After blocking with 5% fat-free milk, the membranes were incubated with the primary antibodies of TLR4, P38, p-P38, IL-6, iNOS, Caspase-3, Bcl-2, GPX4, HO-1, and GAPDH at 4˚C overnight. Next, the membranes were incubated with the secondary antibodies (Santa Cruz, CA; 1:2000) for 1h at room temperature. After the final washes, the protein bands were scanned using the gel imaging system (BIO-RAD, United States).

## Statistical analysis

Data were expressed as the mean ± standard error of the mean (SEM). Statistical significance of differences was assessed by one-way ANOVA followed by Kruskal-Wallis rank sum test for multiple groups and two-tailed, unpaired Student's $t$ test for two groups with the assistance of GraphPad Prism Program (Version 8.2.1). Significance was set at $^*p < 0.05$, $^{**}p < 0.01$, $^{***}p < 0.001$, $^{****}p < 0.0001$.

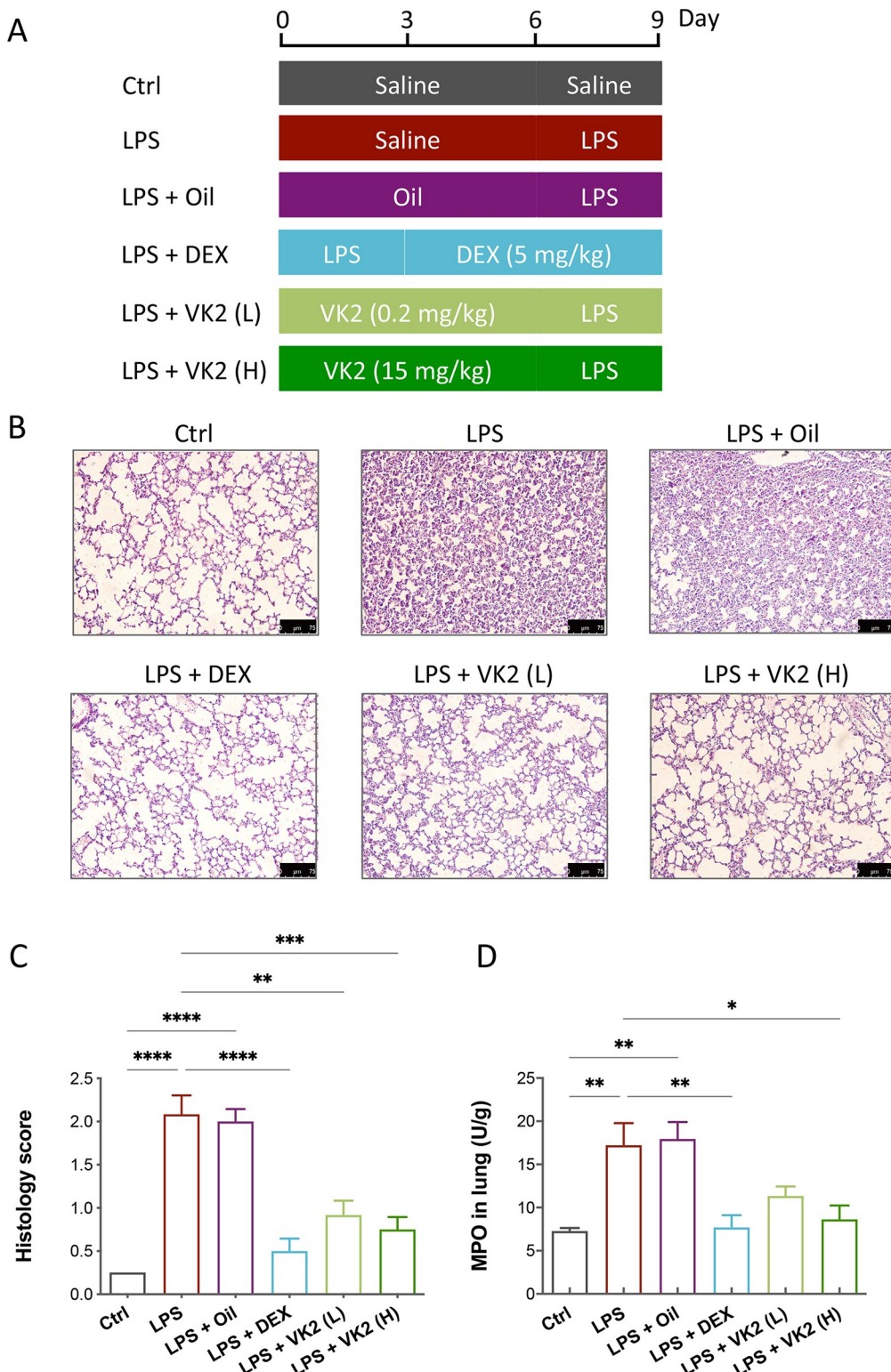

**Fig 1. VK2 attenuated LPS-induced acute lung injury.** (**A**) Mice were pre-administered intragastrically solvent or VK2 (0.2 and 15 mg/kg respectively) and subsequent intraperitoneal injection of LPS (7 mg/kg). (**B**) Histological analysis of lung tissue sections by HE staining (original multiples: 200 ×, scale = 75 μm). (**C**) Lung tissue injury was assessed by histological scores in all groups. (**D**) Determination of the myeloperoxidase (MPO) activity in lung homogenates. Values represent means ± SEM, $^*p < 0.05$, $^{**}p < 0.01$, $^{***}p < 0.001$ and $^{****}p < 0.0001$.

## Results

### VK2 ameliorates LPS-induced histopathological changes and decreases MPO activity in lung

The treatments of mice were shown in **Fig 1A**. We first observed the morphological changes of lung tissues by comparing the HE-stained pathological sections (**Fig 1B and 1C**). LPS-induced mice showed alveoli structure destruction, thickening of alveolar interval, and inflammatory cell infiltration compared with their control counterparts. After VK2 pretreatment, lung injury induced by LPS gradually resolved and followed a dose-dependent trend. Then, the marker of neutrophils infiltration in tissues, MPO, has been detected in this study. The results showed that MPO activity in lung tissues was significantly increased by LPS ($p = 0.0052$). Pretreatment with VK2, especially with high dose, significantly inhibited LPS-induced MPO activity ($p = 0.0175$) (**Fig 1D**). Taken together, these results suggest that VK2 protected against LPS-induced ALI in mice.

### VK2 improved LPS-induced inflammation via P38 MAPK signaling

The results of the content of inflammatory cytokines (TNF-$\alpha$ and IL-6) and anti-inflammatory cytokine (IL-10) in the serum were shown in **Fig 2A–2C**. Compared with control group, contents of TNF-$\alpha$ ($p = 0.0066$, **Fig 2A**) and IL-6 ($p = 0.0014$, **Fig 2B**) were drastically increased, but that of IL-10 ($p = 0.0233$, **Fig 2C**) was decreased in LPS-induced model group. Similar to the positive drug DEX, VK2 administration, especially high dose, displayed the strongest inhibitory effect on reducing TNF-$\alpha$ ($p = 0.0488$) and IL-6 ($p = 0.0116$) and increasing IL-10 ($p = 0.004$). These results indicated that VK2 could alleviate LPS-induced lung inflammation by inhibiting the excessive production of pro-inflammatory cytokines and promoting the production of anti-inflammatory factors.

MAPK is a common signaling pathway in the LPS-induced inflammatory response [21]. The effect of VK2 on MAPK signaling was assessed by measuring the phosphorylation of p38 and the expression of upstream protein (TLR4). Western blot assays showed that compared with the control group, the expressions of TLR4 ($p = 0.0442$, **Fig 2D and 2E**) and phosphory-lated p38 ($p = 0.0009$, **Fig 2D and 2F**) were significantly increased in the LPS group. However, VK2 administration, especially high dose, could obviously inhibited LPS-induced increase of TLR4 ($p = 0.0108$) and phosphorylated p38 ($p = 0.0132$). iNOS and IL-6 are important pro-inflammatory proteins in the cascading inflammatory response, which can be regulated by MAPK. As shown in **Fig 2G–2I**, the expressions of iNOS and IL-6 were increased by LPS compared with control groups, but effectively reduced by VK2 with respect to the LPS model groups ($p < 0.05$). The results suggest that VK2 alleviated LPS-induced inflammation, and the activation and regulation of P38 MAPK pathway was involved in these anti-inflammatory processes.

### VK2 inhibits apoptosis in LPS-induced ALI

In this study, expressions of apoptosis related lung proteins were determined via Western blotting (**Fig 3**). In comparison with the control group, LPS treatment dramatically increased the pro-apoptotic protein expression of Caspase-3 ($p = 0.0077$, **Fig 3A and 3B**), while significantly downregulated the expression of anti-apoptotic protein Bcl-2 ($p = 0.0041$, **Fig 3A and 3C**). However, VK2 pretreatment significantly reduced Caspase-3 and increased Bcl-2 expression levels in ALI mice ($p < 0.05$), suggesting VK2 pretreatment might relieve LPS-induced pulmonary cell apoptosis in ALI mice.

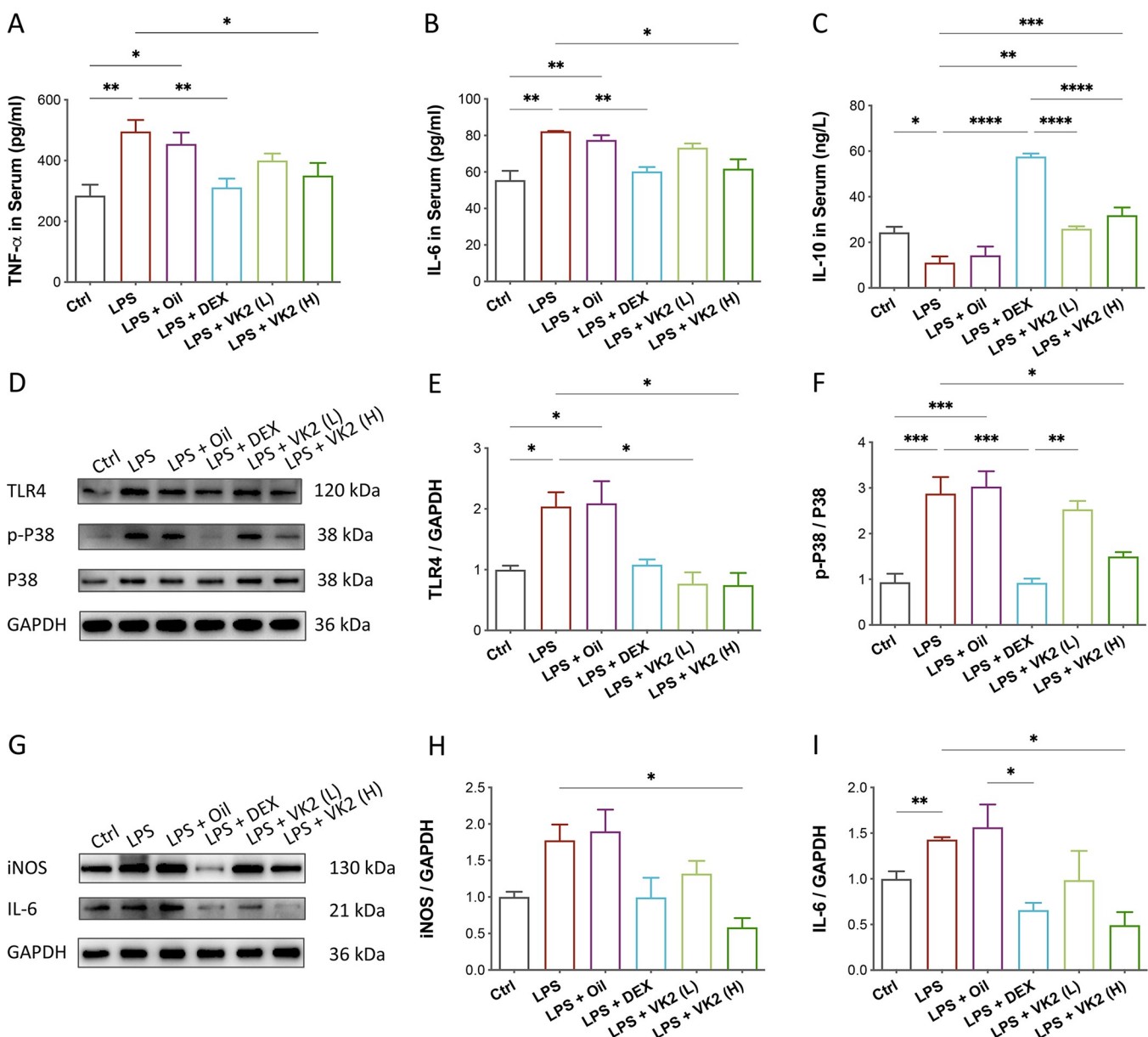

**Fig 2. Effects of VK2 on LPS-induced inflammation.** (**A**) TNF-α, (**B**) IL-6 and (**C**) IL-10 were measured with ELISA in mouse serum. (**D**) The protein expression levels of TLR4, p-P38 MAPK, and P38 MAPK were evaluated by western blotting. (**E**, **F**) Quantitative analysis of TLR4 and the ratio of p-P38/P38 normalized with GAPDH were performed using Image J software. (**G**) The protein expression levels of iNOS and IL-6 were evaluated by western blotting. (**H**, **I**) Quantitative analysis of iNOS and IL-6 normalized with GAPDH. Values represent means ± SEM, *$p < 0.05$, **$p < 0.01$, ***$p < 0.001$ and ****$p < 0.0001$.

## VK2 mitigates ferroptosis in LPS-induced ALI

Ferroptosis results from the iron-dependent accumulation of lipid peroxides. To explore the protective role of VK2 pretreatment against LPS-induced ferroptosis, the level of reductive glutathione (GSH), lipid peroxidation products malondialdehyde (MDA) and tissue iron levels were measured (**Fig 4**). The GSH level was significantly reduced after LPS challenge ($p = 0.0009$), while VK2 pretreatment reversed this situation and increased GSH extent in lung tissue ($p = 0.0462$, **Fig 4A**). In contrast, the MDA level ($p = 0.0015$, **Fig 4B**) and tissue iron ($p = 0.0409$, **Fig 4C**) were significantly decreased after VK2 supplementation in high dose

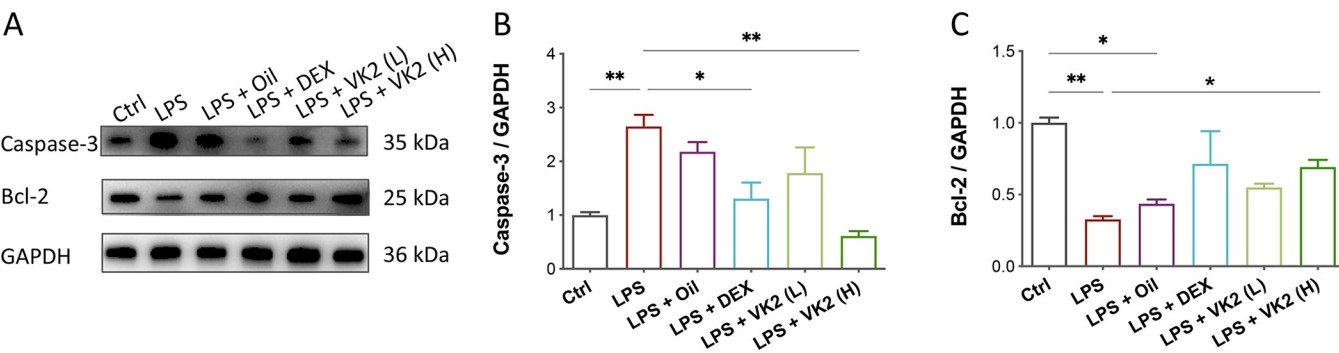

**Fig 3. VK2 inhibits apoptosis in LPS-induced ALI.** (**A**) The protein expression levels of Caspase-3 and Bcl-2 were evaluated by western blotting. (**B**, **C**) Quantitative analysis of Caspase-3 and Bcl-2 normalized with GAPDH were performed using Image J software. Values represent means ± SEM, $^*p < 0.05$ and $^{**}p < 0.01$.

compared with LPS group. Lipid peroxidation is the kernel of ferroptosis, and glutathione peroxidase 4 (GPX4) is an antioxidant enzyme reported as a critical inhibitor of ferroptosis. In LPS-induced mice, GPX4 level decreased drastically ($p = 0.0002$) and could be salvaged by VK2 pretreatment in high dose ($p = 0.0052$) but not positive drug DEX (**Fig 4D and 4E**). Moreover, we found that LPS induction led to a mild elevation of heme oxygenase-1 (HO-1, a stress response protein), while VK2 pretreatment in high dose restored the situation to normal

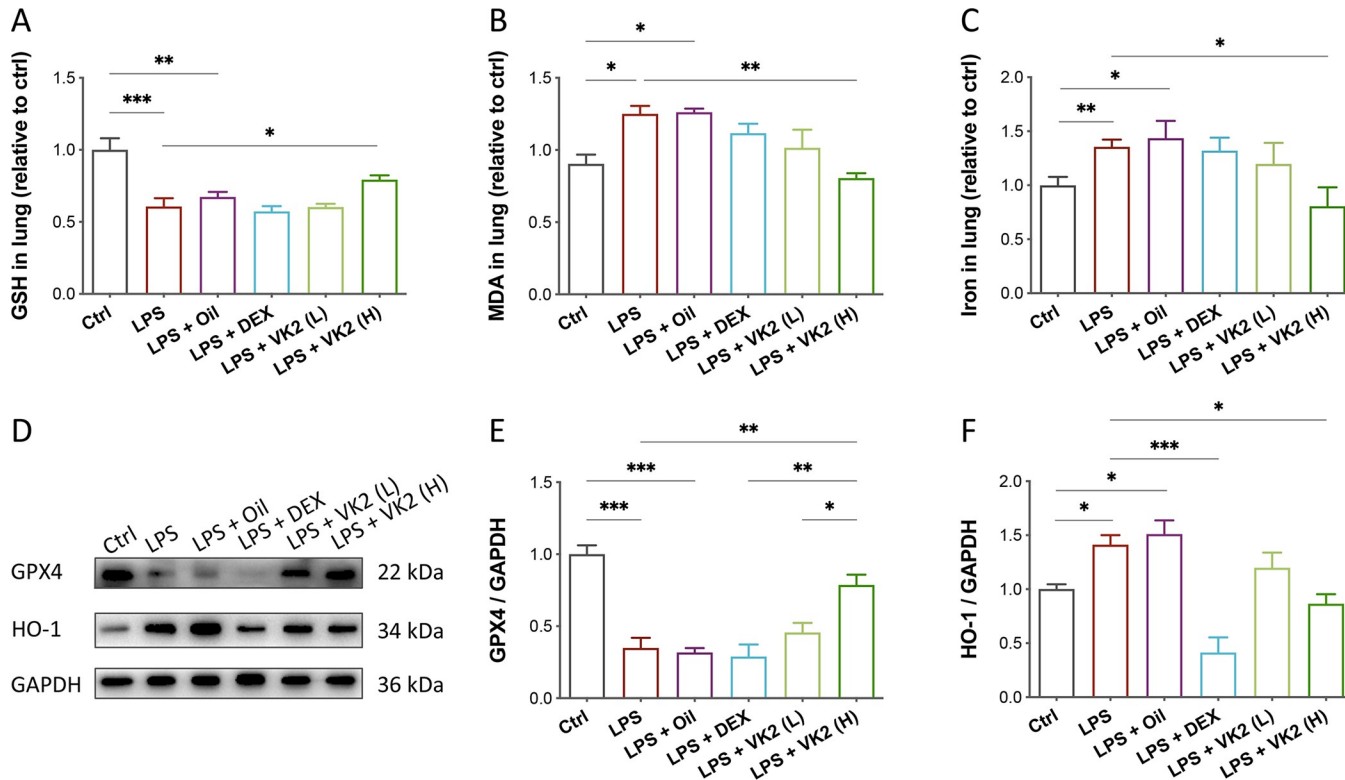

**Fig 4. Role of VK2 in ferroptosis during LPS-mediated injury.** (**A**) GSH, (**B**) MDA, and (**C**) Total iron levels in lung tissues. (**D**) GPX4 and HO-1 protein expression were measured by western blotting. (**E**, **F**) Quantitative analysis of GPX4 and HO-1 normalized with GAPDH were performed using Image J software. Values represent means ± SEM, $^*p < 0.05$, $^{**}p < 0.01$, $^{***}p < 0.001$ and $^{****}p < 0.0001$.

(**Fig 4D and 4F**). Collectively, these results indicated that VK2 was capable of repressing ferroptosis in LPS-induced ALI while DEX was not.

## VK2 inhibits LPS-induced lung elastin degradation

Elastin degradation is implicated in the pathology of ALI [22] and is partially regulated by Matrix Gla Protein (MGP), via a vitamin K-dependent pathway [23]. To evaluate the effect of VK2 pretreatment on LPS-induced lung elastin degradation, the level of uncarboxylated MGP (uc-MGP) and DES (an elastin-specific degradation product) were measured (**Fig 5**). It was found that LPS stimulation significantly increased uc-MGP level ($p = 0.0066$, **Fig 5A**), while VK2 intervention, especially in high dose, reversed this situation and reduced serum uc-MGP level ($p = 0.0053$, **Fig 5A**). However, it is interested that the positive drug DEX could not decrease the serum level of uc-MGP after LPS induction. Moreover, we found that DES level increased drastically in LPS-induced mice ($p = 0.0151$) and could be decreased by VK2 pretreatment in high dose ($p = 0.0044$) an also positive drug DEX ($p = 0.0026$, **Fig 5B**). The above results indicated that VK2 may alleviate lung injury by inhibiting LPS-induced the pulmonary elastin degradation through carboxlation of MGP.

## Discussion

A recent study reported serum VK2 (MK7) in patients with Coronavirus Disease 2019 (COVID-19) was very low compared with non-COVID-19 pneumonia and healthy controls [24]. So does VK2 have a role in prevention and treatment of LPS-induced ALI? Based on our experiment, VK2 could improve LPS-induced ALI. We determined the VK2 intervention dose for this experiment based on previous studies [18, 19, 25] and on the range of recommended VK2 intakes in each country [26, 27].

The inflammatory response is an important defense mechanism induced in the host in response to injury, infection or stimulation [28]. Under LPS induction, the levels of proinflammatory cytokines, including interferons (IFNs), tumor necrosis factors (TNFs), interleukins (ILs), and chemokines [29] were significantly higher than the control group, and the infiltration of inflammatory cells in lung tissue was upregulated as well, which was consistent with

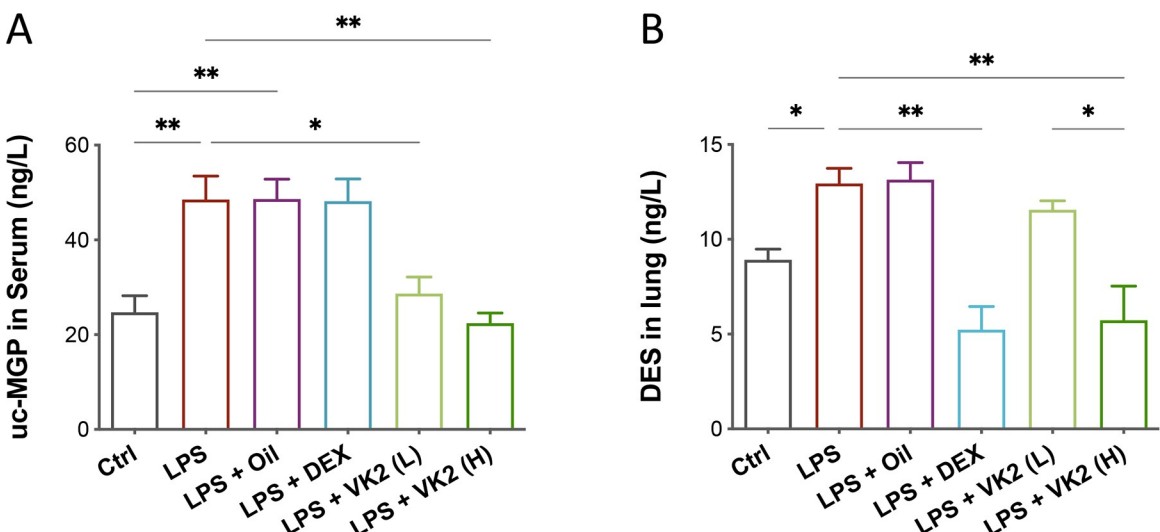

**Fig 5. VK2 alleviates LPS-induced lung elastin degradation.** (A) uc-MGP levels in serum. (B) DES levels in lung tissues. Values represent means ± SEM, *$p < 0.05$ and **$p < 0.01$.

our results. However, VK2 pretreatment significantly improved these phenomena and gradually returned to normal levels. VK2 has been investigated as a potential anti-inflammatory and protective drug in several inflammatory diseases including type 2 diabetes mellitus (T2DM) [16], inflammatory bowel disease (IBD) [25], atherosclerosis [30], rheumatoid arthritis (RA) [31], and neurodegenerative diseases, especially Parkinson's and Alzheimer's disease [32, 33].

Mitogen-activated protein kinases (MAPKs) are a class of serine/threonine protein kinases in cells, which mediate cellular responses to extracellular stimulations, including growth factors, cytokines, hormones, neurotransmitters, cellular stress, and cell adherence [34]. Activation of the MAPK signaling pathway is also involved in the occurrence of inflammation in LPS-induced ALI [35]. Further researches suggested that the P38 MAPK pathway played a crucial role in LPS-stimulated inflammatory response and macrophage activation [36]. In this study, we found that in the LPS model group, VK2 significantly reduced the proportion of p-P38/P38, thus reducing the expression of inflammatory cytokines IL-6 and iNOS. Therefore, these data suggested that VK2 might alleviate inflammation by inhibiting the activation of the P38 MAPK pathway.

Apoptosis is the process of programmed cell death that occurs under normal physiological or pathological conditions [37]. Excessive apoptosis plays an important role in the development of ALI [38, 39]. LPS can promote increased cell apoptosis by triggering inflammatory responses [40]. Bcl-2 controls the cell proliferation or apoptosis through inhibiting cell apoptosis and the activation of downstream Caspase-3 proteases [41]. The results of this study indicated that VK2 exerted its protective effect on ALI through the upregulation of Bcl-2 and downregulation of Caspase-3 in LPS-induced ALI. Indeed, VK2 administration has been reported to exert anti-apoptotic and anti-inflammatory effects [18, 42]. Furthermore, P38 MAPK pathway has been evidenced to mediate apoptosis [43]. Studies have shown that phosphorylated P38 MAPK can activate caspase-3 and promote apoptosis by downregulating Bcl-2 [44–46]. Thereby, VK2 reduced lung cell apoptosis by inhibiting LPS-induced activation of P38 MAPK.

Recent studies have found that ferroptosis also has a crucial function in the progression of ALI, and its inhibition is effective in alleviating ALI [47–50]. Ferroptosis is a unique iron-dependent lipid-peroxidation, which is different from traditional necrosis, apoptosis, autophagy, or other forms of cell death [51]. It is mainly characterized by the accumulation of free iron, drastic lipid peroxidation, and ROS production [52]. According to the occurrence process of ferroptosis, various key factors including GSH, MDA, iron, and GPX4 are often used to comprehensively evaluate lipid peroxidation and ferroptosis [53]. In this study, we also found significant ferroptosis in LPS-induced ALI, characterized by increased MDA and iron levels, and reduced GSH and GPX4 levels. After VK2 administration, the accumulations of MDA and iron were significantly reduced while the expression of GPX4 was obviously increased, suggesting that VK2 could inhibit the LPS-induced ferroptosis. Heme oxygenase 1 protein (HO-1), an important stress response protein highly expressed in lungs, is encoded by HMOX1 gene and decomposes heme into iron, carbon monoxide (CO), biliverdin, and bilirubin, which is essential in the balance of intracellular iron and ROS [54]. HO-1-deficient mice exhibit high oxidative damage, tissue injury, and chronic inflammation, as well as hepatic and renal iron accumulation [55]. However, HO-1 has not only beneficial role to protect against oxidative stress but also detrimental role to promote ROS generation by releasing free ferrous iron and subsequent expression of ferritin [56–59]. Moreover, HO-1-induced ferroptosis may be related to the reduction of free iron-binding ability of ferritin induced by oxidative stress, such as iron accumulation and lipid peroxidation [60]. In this study, we found that administration of VK2 effectively inhibited LPS-induced elevation of HO-1. According to these data, it

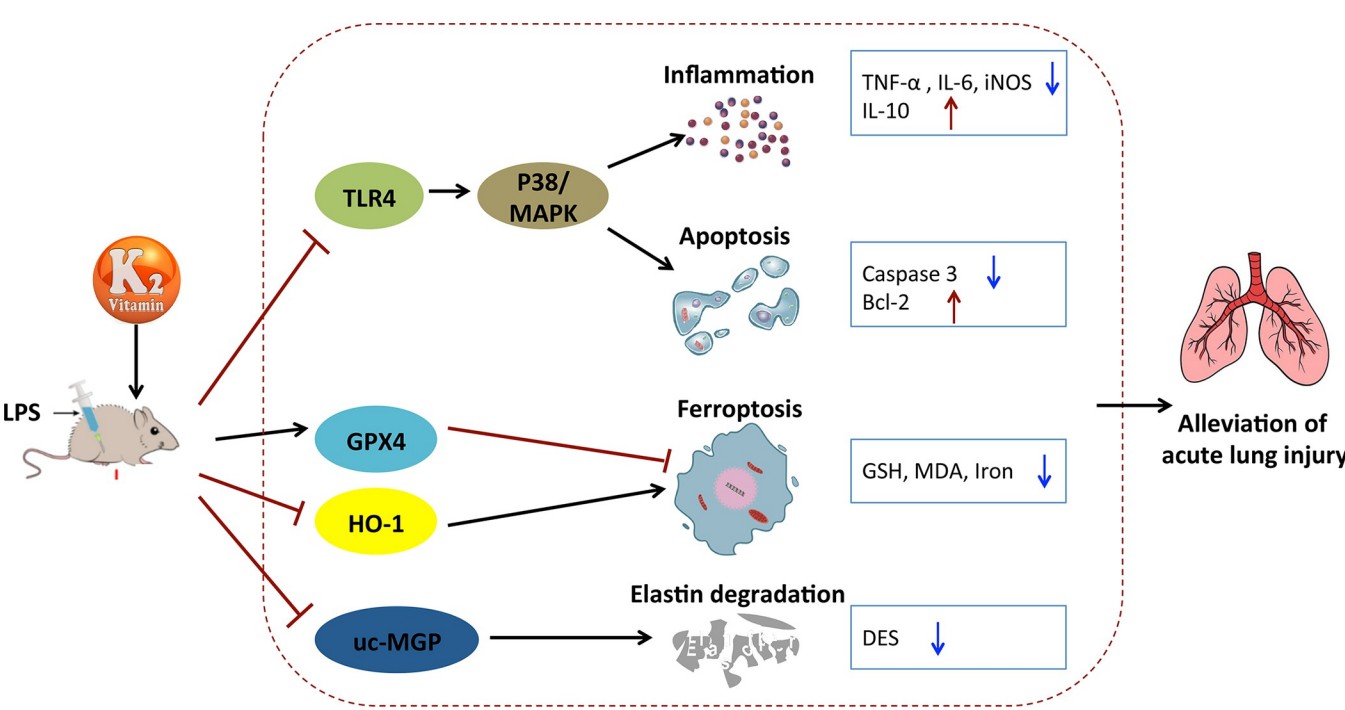

**Fig 6. Schematic representation of VK2 action on LPS-induced ALI.**

can be inferred that the protective effect of VK2 on LPS-induced lung injury might be related to ferroptosis.

Elastic fibers are important components of the extracellular matrix of dynamic tissue, conferring resilience of the lungs and arteries, subsequently promoting breathing and circulation [61]. DES is formed during the crosslinking process of nourishing layer-elastin polymers, which is a product of elastin degradation, releasing into the blood after elastic fibers degradation [62]. MGP has an important role in elastic fibers degradation and requires VK for its activation [63]. Uncarboxylated MGP (uc-MGP) is a biomarker of the VK status in the body, and high level of uc-MGP reflect the low vitamin K status [64]. In current study, we found that uc-MGP concentration was significantly increased by LPS, while after VK2 pretreatment, it was significantly decreased, which was consistent with the results of DES. These results suggested that VK2 supplementation could carboxylate uc-MGP into MGP, effectively preventing the degradation of elastic fibers and protecting the lungs.

In summary, the findings of the current study suggest that VK2 effectively protects against ALI caused by LPS through suppression of P38 MAPK signaling pathway and ferroptosis as depicted in **Fig 6**. Our work suggested that VK2 supplementation might be a cheap and effective intervention or prevention measures against serious courses of ALI. Nevertheless, despite widespread adoption, the LPS-induced ALI model may not entirely encompass the complexity and heterogeneity of human ALI. It is essential to bolster animal studies with clinical research and validation. In the future, we will carry out relevant in-depth work to solve this problem.

## Supporting information

**S1 Raw images.**
(ZIP)

## Author Contributions

**Conceptualization:** Lihong Liu, Jiepeng Chen, Lili Duan, Yuyuan Li, Shuzhuang Li.

**Data curation:** Yulian Wang, Weidong Yang, Lulu Liu.

**Formal analysis:** Yuyuan Li.

**Funding acquisition:** Yuyuan Li, Shuzhuang Li.

**Investigation:** Yulian Wang, Weidong Yang, Jiepeng Chen, Lili Duan, Yuyuan Li, Shuzhuang Li.

**Project administration:** Yulian Wang, Weidong Yang, Lulu Liu.

**Software:** Lihong Liu, Jiepeng Chen, Lili Duan.

**Supervision:** Lihong Liu, Shuzhuang Li.

**Validation:** Yulian Wang, Weidong Yang, Lulu Liu.

**Writing – original draft:** Yuyuan Li.

**Writing – review & editing:** Lihong Liu, Shuzhuang Li.

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
