## [Decision Letter · Decision Letter 0]

7 Sep 2023

PONE-D-23-23562Vitamin K2 (MK-7) attenuates LPS-induced acute lung injury via inhibiting inflammation, apoptosis, and ferroptosisPLOS ONE

Dear Dr. Li,

Thank you for submitting your manuscript to PLOS ONE. After careful consideration, we feel that it has merit but does not fully meet PLOS ONE’s publication criteria as it currently stands. Therefore, we invite you to submit a revised version of the manuscript that addresses the points raised during the review process.

We look forward to receiving your revised manuscript.

Kind regards,

Mohamed Ezzat Abd El-Hack

Academic Editor

PLOS ONE

- https://www.frontiersin.org/articles/10.3389/fphar.2021.790072/full

- https://www.mdpi.com/2072-6643/13/12/4441/htm

- https://pubmed.ncbi.nlm.nih.gov/36002086/

In your revision ensure you cite all your sources (including your own works), and quote or rephrase any duplicated text outside the methods section. Further consideration is dependent on these concerns being addressed.

3.PLOS ONE now requires that authors provide the original uncropped and unadjusted images underlying all blot or gel results reported in a submission’s figures or Supporting Information files. This policy and the journal’s other requirements for blot/gel reporting and figure preparation are described in detail at https://journals.plos.org/plosone/s/figures#loc-blot-and-gel-reporting-requirements and https://journals.plos.org/plosone/s/figures#loc-preparing-figures-from-image-files. When you submit your revised manuscript, please ensure that your figures adhere fully to these guidelines and provide the original underlying images for all blot or gel data reported in your submission. See the following link for instructions on providing the original image data: https://journals.plos.org/plosone/s/figures#loc-original-images-for-blots-and-gels. 

Reviewers' comments:

Reviewer's Responses to Questions

**Comments to the Author**

1. Is the manuscript technically sound, and do the data support the conclusions?

Reviewer #1: No

Reviewer #2: Yes

2. Has the statistical analysis been performed appropriately and rigorously? 

Reviewer #1: Yes

Reviewer #2: Yes

3. Have the authors made all data underlying the findings in their manuscript fully available?

Reviewer #1: No

Reviewer #2: Yes

4. Is the manuscript presented in an intelligible fashion and written in standard English?

Reviewer #1: Yes

Reviewer #2: Yes

5. Review Comments to the Author

Reviewer #1: Comments to PONE-D-23-23562

As we all know, the COVID-19 pandemic is a research hotspot. Because of this, many researchers are trying to grasp this hotspot. But a lot of the research has nothing to do with COVID-19. The current study can only be said to focus on the relationship between Vitamin K2 and LPS-induced acute lung injury, and the relationship with COVID-19 is minimal. However, the authors introduce too much COVID-19 in the Abstract and Introduction. The authors must be aware of this and make the necessary corrections. There is no need for researchers to rub hot spots.

Lipopolysaccharide (LPS) can indeed cause acute lung injury (ALI) or acute respiratory distress syndrome (ARDS). The symptoms of ALI induced by LPS and those of a new coronavirus infection may share some similarities, but there are important distinctions between the two conditions.

Some of my specific comments can be found below:

1. Lines 76 and 77: This seemingly straightforward sentence is an essential basis for your research, but unfortunately, this reference cannot support this argument. Please carefully check this citation.

2. Line 131: How did the authors choose this exposure dose? Did the mice die at this dose? What is the mortality rate?

3. Lines 269 to 279: Please see my above comments.

4. Lines 351 to 353: This reviewer does not think this sentence could be listed in the Conclusion Section. The authors should never forget that you only injected LPS intraperitoneally into mice and did not artificially infect experimental animals with the new coronavirus.

Reviewer #2: Due to its multiple pharmacological properties, VK2 has become a research hotspot for scientists in recent years, including anti-inflammatory, anti-oxidant, anti-apoptotic, and anti-ferroptosis properties. According to this study, VK2 protects against ALI caused by LPS by suppressing the P38 MAPK signaling pathway and ferroptosis. Overall, the paper is well-written, but a few minor revisions are necessary.

1.     Some abbreviations that appeared in the abstract were not explained, such as "MDA" and "GSH".

2.     In animal experiments, why is it that the positive control group receives the LPS injection first, while the other groups receive the injection afterward? It is necessary to provide a reasonable explanation for this. Would it be better to replace a positive control group?

3.     Is it necessary to replace "treatment" with "pretreatment" in line 239, line 246 and line 345?

4.     As shown in Figures 4 A, B, and C, the results are not significant and require further investigation.

5.     The association between P38 MAPK pathway and Bcl-2 and Caspase-3 needs to be explained in lines 308-310.

6.     The format of the article is not standardized, such as ' Caspase-3 ' in several places without capitalization. It is strongly recommended that you double-check.

7.     The limitations of this experiment were not discussed in the discussion.

6. PLOS authors have the option to publish the peer review history of their article (what does this mean?). If published, this will include your full peer review and any attached files.

Reviewer #1: **Yes: **Fan Yang

Reviewer #2: **Yes: **Li Deng

---

## [Author Response · Author response to Decision Letter 0]

19 Sep 2023

Dear Editor: 

First, I would like to express my deep gratitude to you and to the reviewers for the constructive comments and suggestions. We have completely revised the manuscript and addressed all the concerns raised by the reviewers.

A Point-by-Point Response to the editor’s and reviewers’ comments is given in the following pages. We hope that the revised manuscript can be accepted by PLOS ONE.

Sincerely

Yuyuan Li, Ph.D.

E-mail: liyuyuan831221@163.com

Advanced Institute for Medical Sciences, Dalian Medical University

Tel.: +86 411 8611 8983.

---

## [Decision Letter · Decision Letter 1]

9 Nov 2023

Vitamin K2 (MK-7) attenuates LPS-induced acute lung injury via inhibiting inflammation, apoptosis, and ferroptosis

PONE-D-23-23562R1

Dear Dr. Li,

We’re pleased to inform you that your manuscript has been judged scientifically suitable for publication and will be formally accepted for publication once it meets all outstanding technical requirements.

Kind regards,

Keiko Hosohata, Ph.D.

Academic Editor

PLOS ONE

Reviewers' comments:

Reviewer's Responses to Questions

**Comments to the Author**

1. If the authors have adequately addressed your comments raised in a previous round of review and you feel that this manuscript is now acceptable for publication, you may indicate that here to bypass the “Comments to the Author” section, enter your conflict of interest statement in the “Confidential to Editor” section, and submit your "Accept" recommendation.

Reviewer #1: All comments have been addressed

Reviewer #2: All comments have been addressed

2. Is the manuscript technically sound, and do the data support the conclusions?

Reviewer #1: Yes

Reviewer #2: Yes

3. Has the statistical analysis been performed appropriately and rigorously? 

Reviewer #1: Yes

Reviewer #2: Yes

4. Have the authors made all data underlying the findings in their manuscript fully available?

Reviewer #1: Yes

Reviewer #2: Yes

5. Is the manuscript presented in an intelligible fashion and written in standard English?

Reviewer #1: (No Response)

Reviewer #2: Yes

6. Review Comments to the Author

Reviewer #1: The author has fully addressed my previous comments. I think this manuscript is acceptable in its current state.

Reviewer #2: (No Response)

7. PLOS authors have the option to publish the peer review history of their article (what does this mean?). If published, this will include your full peer review and any attached files.

Reviewer #1: **Yes: **Fan Yang

Reviewer #2: **Yes: **Li Deng

---

## [Editor Report · Acceptance letter]

15 Nov 2023

PONE-D-23-23562R1 

Vitamin K2 (MK-7) attenuates LPS-induced acute lung injury via inhibiting inflammation, apoptosis, and ferroptosis 

Dear Dr. Li:

I'm pleased to inform you that your manuscript has been deemed suitable for publication in PLOS ONE. Congratulations! Your manuscript is now with our production department. 

Kind regards, 

on behalf of

Dr Keiko Hosohata 

Academic Editor

PLOS ONE